# Effects of Coix Seed Oil on High Fat Diet-Induced Obesity and Dyslipidemia

**DOI:** 10.3390/foods11203267

**Published:** 2022-10-20

**Authors:** Lichun Chen, Songwen Xue, Binhao Dai, Huimin Zhao

**Affiliations:** Food Safety Key Laboratory of Zhejiang Province, School of Food Science and Biotechnology, Zhejiang Gongshang University, Hangzhou 310018, China

**Keywords:** coix seed oil, fatty acid, lipid metabolism, inflammatory factors, serum lipid levels, high-fat diet

## Abstract

Dietary intervention is becoming more popular as a way to improve lipid metabolism and reduce the prevalence of diet-related chronic disorders. We evaluated the effects of several dietary oils on body weight, fat mass, liver weight, and tumor necrosis factor in obese mice given a high-fat diet (HFD) to discover if coix seed oil (CSO) had an anti-obesity impact. As compared to other dietary fats, CSO treatment considerably lowered body weight and liver index, successfully sup-pressed total cholesterol and triglyceride content, and raised liver lipid deposition and lipid metabolism problem induced by high fat intake. Furthermore, gas chromatography research revealed that CSO extracted by supercritical fluid, with 64% being CSO extracted by supercritical fluid, and the greatest amounts of capric acids and lauric acids being 35.28% and 22.21%, respectively. CSO contained a high content of medium-chain fatty acids and was able to modify hepatic fatty acid metabolism and lipid levels in HFD-induced obese mice. According to the results, CSO has the potential to replace dietary lipids as a promising functional lipid in the prevention of met-abolish disorders.

## 1. Introduction

More and more scientists are concerned about chronic diseases caused by daily diet, especially obesity-related hyperlipidemia, diabetes, and fatty liver. Hope to intervene in diseases related to lipid metabolism disorders through dietary intervention [1,2,3,4]. The lipids in food are esterified with the apolipoprotein synthesized on the rough endoplasmic reticulum after esterification on the smooth endoplasmic reticulum [5], thereby, activating lipoprotein lipase to catalyze the hydrolysis of triglyceride to fatty acids and glycerol. Studies have shown that obesity and lipid metabolism disorders are the excessive proliferation and differentiation of adipocytes leading to the formation of poly adipocytes [6,7]. This might explain why adipose tissue not only stores energy, but also secretes adipocytokines including leptin, tumor necrosis factor-α (TNF-α), and interleukin-β (IL-β) to maintain the balance of various physiological activities in the body [8,9]. For example, peroxisome proliferator-activated receptor (PPAR-γ) was a type of ligand-activated nuclear transcription factors. Peroxisome proliferator-activated receptor (PPAR-γ) was highly expressed in adipocytes and was important regulators of adipocyte gene expression and insulin cell signaling [10]. It has a specificity of adipose tissue and requires for adipocyte development. Some observations in vitro and in vivo indicate that PPAR-γ can inhibit oxidative stress through transcriptional activation of antioxidant enzymes such as superoxide dismutase and catalase [11]. In addition, PPAR-γ can be adjusted by altering the endocrine function of adipose tissue (such as leptin, IL-6, TNF-α, etc.). Previous studies have shown that there may be potentially associated between lipid components and the risk of inflammation [12,13]. Functional factors derived from food or dietary supplements play an important role in promoting health and preventing disease. Recent changes in consumer lifestyles have shifted demand towards safe and quality foods, such as dietary oils that improve lipid metabolism [14,15].

Coix (Coix *lachrymal-jobi* L.), commonly known as “adlay” [16], is a potential food source, containing a variety of bioactive components, such as polysaccharides [17], polyphenols [18], coixenolide [19,20], and cortisol [21]. Some studies have shown that coix seed extracts have prebiotic potential, increasing the accumulation of bioactive metabolites such as phenyl lactate, vitamins and biotin [22]. Tanimura’s group discovered the active molecule with anticancer properties to be an oil component, coixenolide [23]. In addition, CSO can lead to apoptosis of cancer cells by modulating mitochondrial functional impairment [24]. In addition to coixenolide, other oil components, including oleic acid and linoleic acid [25,26], have also been shown to inhibit tumor growth in transplanted mice [27,28]. Among them, unsaturated fatty acids (UFAs) with the distinct advantages of good biocompatibility and innate tumor-targeting effect [29], have been widely researched for chemotherapeutic agent-unsaturated fatty acid prodrugs. CSO has been shown to prolongs lifespan and enhances stress resistance in Caenorhabditis elegans [30]. In recent years, a growing number of phytochemicals were investigated to lower blood lipid in the obese mice and ameliorated hyperlipidemic syndrome. Several studies have shown that supplementation with coix seeds affects dietary fat-induced metabolic syndromes and nonalcoholic steatohepatitis [31]. In our previous studies, we have found that polysaccharides from coix seed showed hypoglycemic properties and alleviated diabetic complications in diabetic mice [32,33]. Moreover, nothing was known about the relationship between long-term CSO use and dyslipidemia and immune system regulation. In particular, the effects of FAs chain length and unsaturation on the CSO have not been well studied.

The purpose of this study was to investigate the effects of different kinds of edible oils, particularly CSO, on lipid metabolism disorder and inflammation in HFD mice. Therefore, we analyzed the physicochemical properties of different oils, as well as the composition and distribution of fatty acids. Moreover, we examined the effects of CSO treatment on TNF-α expression, body weight changes, serum triglyceride concentrations and total cholesterol in the adipose tissue of obese HFD mice. The study in this paper contributes to our understanding of the theoretical basis regarding the improvement of lipid metabolism and prevention of obesity by CSO.

## 2. Experimental

### 2.1. Materials

Coix seed oil was extracted by Kanglaite Co., Ltd. (Hangzhou, China), and other edible oils were supplied from the local supermarket (Hangzhou, China). Commercial assay kits such as triglyceride (TG, Number: A110-1-1), serum total cholesterol (TC, Number: A111-1-1), aspartate transaminase (AST, Number: C010-2-1) and alanine transaminase (ALT, Number: C009-2-1) were obtained from Jiancheng of Bioengineering Institute (Nanjing, China). The monoclonal antibodies for β-actin (Number: A1978), IL-6 (Number: SRP3330), TNF-α (Number: 654225), and IL-1β (Number: RAB0274) were provided by Merck-Millipore (Shanghai, China). Hematoxylin-Eosin (H&E) staining kit (Number: G1120) was purchased Solarrbio Technology Co., Ltd. (Beijing, China). All other reagents were used to analyze the highest commercial grade available (they were provided by Merck, Shanghai, China).

### 2.2. Fatty Acid Composition of Coix Seed Oil

We used cold press technology to obtain CSO. The oil was hydrolyzed with sn-1,3-specific pancreatic lipase (Sigma-Aldrich, Shanghai, China) to determine the fatty acid composition at the sn-2 position, as previously described [34]. Briefly, pancreatic lipase (from porcine pancreas, 20 mg), 1 mL of 1 M Tris-HCl buffer (pH 7.6), 0.25 mL of 0.05% bile salts, 0.1 mL of 2.2% calcium chloride were added to a test tube containing 100 mg oil. The mixture was vortexed for 3 min while being incubated at 40 °C in a water bath. Then HCl solution (6 M, 1 mL) and ether (1 mL) were added to stop the enzyme reaction. The organic phase was eluted in a column with anhydrous sodium sulfate. Finally, the methylation reaction was carried out by adding 0.5 mol/L methanol sodium hydroxide solution and 14% BF_3_-methanol solution and heating in a boiling water bath for 10 min. After cooling, isooctane and saturated sodium chloride solution were added and separated through a column of anhydrous sodium sulfate.

The fatty acid composition was analyzed by gas chromatography (GC), using an Agilent 6890 instrument (Palo Alto, Santa Clara, CA, USA) equipped with a flame ionization detector (FID), 100 m × 0.25 mm column (CP-Si188, Chrompack, Carlsbad, CA, USA). The sample was maintained at 60 °C for 1 min after injection, then the temperature was raised to 130 °C with a rate of 10 °C/min at, increased to 170 °C at a rate of 3 °C/min, 210 °C at a rate of 5 °C/min and finally to 230 °C at 10 °C/min, holding for 5 min. Hydrogen was used as a carrier gas, the column inlet pressure setting at 20 psi and a split ratio of 1:20. We calculated the fatty acid composition at the sn-1,3 position using the following formula [35]:$$FAc_{\mathrm{sn}-1,3}\left(\%\right)=\frac{\left[\left(3\times TotalFAc\right)-FAc_{\mathrm{sn}-2}\right]\;}{2}$$

### 2.3. Experimental Design and Treatment

Male ICR mice (25 ± 1.5 g; 6 weeks old) were purchased from SLAC Co., Ltd. (Shanghai, China). The mice were housed in individual stainless-steel cages (5/cage) and kept in a standard controlled environment (25–27 °C, 50 ± 10% relative humidity with a light/dark cycle for 12 h). The animals were acclimated for one week before being used and allowed free access to a standard chow diet and water. Before the experiment, all animals were given a high-fat diet (from SLAC, Shanghai, China) with a total calorie of 4.11 kcal/g for twenty-six weeks, and until they met the obesity model standard as described by previous studies [36,37]. Then, the HFD mice were randomly divided into five experimental groups (*n* = 15 of each group): the control group (CON) was given normal saline; the peanut oil group (PE) and lard group (LA) received 30 mg/kg/BW oil; and the coix seed oil groups received 30 mg/kg/BW (CSO-1) and 50 mg/kg/BW (CSO-2) by oral injection CSO, once a day, continuing 5 weeks. body weight and food intake were monitored during the initial adaptation period and experimental treatment. The control group (CON) received normal saline instead of oil. After experimental administration, the mice fasted overnight, and serum samples were separated from orbital eye bleeding under anesthesia. Then the different tissue sections were weighed and stored at −80 °C to prepare for subsequent experiments. 

### 2.4. Biochemical Assays and Tissue Parameters

Liver tissues were dissolved in cold saline at the ratio of 1:10 (*w:v*), then centrifuged at 3000 r/min lasting 10 min to get supernatant, for ADH, ALDH, SOD, MDA, and protein to test (ADH: Alcohol dehydrogenase assay kit, Number: A083-2-1; ALDH: Aldehyde dehydrogenase assay kit, Number: A075-1-1; SOD: Total Superoxide Dismutase assay kit, Number: A001-1-1; MDA: Malondialdehyde assay kit, Number. They are all provided by Jiancheng of Bioengineering Institute, Nanjing, China). The serum AST and ALT activities were detected by using assay kits according to the instructions, and the concentration of serum IL-1β and TNF-α were detected by using commercial ELISA kits (Number: RAB0335, Merck-Millipore, MN, USA) following the instructions [38].

### 2.5. Histopathology

The H&E staining was performed as previously described [39]. Briefly, liver tissues were isolated and a part of the tissue was fixed with 5% polyoxymethylene. Then samples were embedded in paraffin for staining with hematoxylin & eosin (H&E). Dehydrated samples were mounted using DPX Mountant (Sigma-Aldrich) and were photographed by DM3000 light microscopy (Leica, Munich, Germany).

### 2.6. Isolation of Total Proteins and Western Blotting

According to the manufacturer’s agreement for users (Roche, Basel, Switzerland), the protein contents in the tissues were extracted with RIPA solution [40]. After being separated on an SDS-PAGE gel, the protein was transferred to polyvinylidene difluoride membranes (Millipore, Billerica, MA, USA). Protein content was determined using the bicinchoninic acid (BCA) protein assay kits. After blocking, the cell membrane was detected with β-Actin, IL-6, TNF-α, and IL-1β antibodies at 4 °C during the night. Following three washes, the secondary antibody (1:5000) and membranes were incubated together for 1 h, and according to the manufacturers’ agreement, the antibody bindings were revealed by ECL Plus Western blotting detection system. The picture was taken by using the imaging system (ChemiDoc XRS+, BIO-RAD, Mannheim, Germany).

### 2.7. Statistical Evaluation 

Data processing was performed using the statistical software SPSS (SPSS Inc., Chicago, IL, USA), and one-way analysis of variance (ANOVA) was used for comparison among groups. All data were reported as mean ± SEM, and statistical significance was considered at *p* < 0.05 (*), *p* < 0.01 (**).

## 3. Results 

### 3.1. Fatty Acid Composition and Triacylglycerol Species

Triacylglycerols were the most abundant lipids in the oils, accounting for more than 95% of the total oil mass. The length of the fatty acid chain and positional distribution in the triacylglycerol molecule determine the physicochemical properties of oils. The area normalization technique was used to compute the levels of total fatty acids and sn-2 fatty acids, and random distribution theory was used to examine the composition and content of sn-1,3 fatty acids. The total fatty acid composition in oil was presented in Table 1. The GC analysis showed that CSO was dominated by saturated medium-chain fatty acids (area%), with capric acids and lauric acids having the highest content, 35.28% and 22.21% respectively. CSO was mainly composed of saturated fatty acids, with a total amount as high as 64%. They were mostly made up of medium chain fatty acids (MC-FA, C8-C12) and long chain fatty acids (lc-fa, c14-c24), accounting for 60.9% and 12.77% of the total. At the sn-2 position, The CSO had a significantly higher content of MC-FA (27.17%) and lower content of LC-FA (4.69%) than others, whereas the levels of sn-2 MC-FA in PE (11.78%) were comparable with those in LA (8.66%).

Because CSO extracted by supercritical fluid has different fatty acid structures. It has also been reported that the fatty acids which located at the sn-2 position were more resistant to autoxidation and had superior bioavailability than those distributed in the sn-1 or sn-3 positions [41,42].

### 3.2. Effects of CSO on Body and Liver Weight

The effects of dietary oil on body weight were shown in Figure 1A, there was no significant difference in body weight between groups at the initial. At the end of the experiment, the body weight had an obvious increase from 24.60 ± 2.15 g to 44.60 ± 2.15 g and 47.35 ± 4.28 g because of the peanut oil and lard treatment, respectively. The energy conversion rate of lard was the highest among the three groups. The growth value of the CSO groups was significantly lower than the CON group’ s, and their average body weights were 43.93 ± 2.96 g and 41.93 ± 1.96 g, respectively. The log-rank test was used to analyze the changes in liver weight (Figure 1B) and fat weight (Figure 1C), and the results showed that there was a significant difference between the CON group and the edible oil group (*p* < 0.01). The peanut and lard treatments increased the liver index by 0.0423 ± 0.011 and 0.0443 ± 0.012, respectively (*p* < 0.01). The liver index values of the CSO groups had no significant difference from that of the control group. Compared with the PE and LA group, CSO treatment reduced the liver weight of the mice, and the livers were less fatty. Moreover, the fat index of the mice in the PE and LA groups was significantly higher than the control group. However, this number was decreased by the CSO treatment, indicating CSO interfered with fat formation. Compared with lard and peanut oil, CSO had a significant impact on reducing the fat index of mice. The results showed that reducing abdominal fat deposition had a significant effect on the occurrence of fatty liver and the control of weight gain.

### 3.3. Effect of CSO on Biochemical Alterations of Serum

The fat entered the liver through lipid metabolism, continuously synthesizing into TG, and entered the blood circulation in the form of lipoproteins (mainly LDL). As shown in Figure 2A, the serum total cholesterol (TC) values of the control group, PE group, and LA group were 5.23 ± 0.41 mM, 6.15 ± 0.68 mM, and 6.43 ± 0.53 mM (*p* < 0.05). The CSO group was considerably lower than the other two groups (*p* < 0.01). Triglycerides and low-density lipoprotein cholesterol (LDL-C), both of which are present in the blood as cholesterol and lipoproteins, can cause various types of physical damage. The TC is a kind of fat in the blood that can be used as a lipid metabolism quota. In this experiment, the CSO administration partly reduced the blood TG concentration, and the LDL-C level of the CSO group was significantly lower than that of the PE and LA groups (*p* < 0.01). The TC/HDL-C level (1.42 ± 0.07) of the CSO group was significantly lower than that of the lard group and other groups. The above experiment results certificated that CSO could improve ameliorating hyperlipidemia. The liver was crucial for the synthesis and metabolism of endogenous blood lipids and lipoproteins. Once the intrahepatic synthesized lipids and blood lipids were unbalanced, TG would be accumulated in the liver to form non-alcoholic fatty liver. The alanine aminotransferase (ALT) and aspartate aminotransferase (AST), enzymes required for the conversion of carbohydrates and proteins in the human body, were two sensitive indicators of hepatocyte injury and the most important parameters in routine clinical liver tests. The results showed that the serum ALT and AST of the CSO group were remarkably lower than those of the lard and peanut oil groups. As shown in Figure 2B,C, after administration of lard or peanut oil, the serum ALT and AST values of the mice were higher than those of the CON group. The findings confirmed that, CSO seed oil had a preventive effect on liver damage caused by lipid metabolism disorders. MC-FAs could be hydrolyzed by gastric lipase, absorbed in the intestine, and then delivered to the liver through the portal vein, where they were consumed as a fast energy source [43,44].

### 3.4. Effect of CSO on Oxidative Stress and Inflammation 

A high-fat diet not only causes the disorder of lipid metabolism but also activates the oxidation pathway of nicotinamide adenine dinucleotide phosphate, generating more reactive oxygen species (ROS) and causing oxidative stress. According to research, the accumulation of lipid peroxidation products reduces the biological activity of cells, leading to dysfunctions such as energy metabolism and cell signal transduction [45]. The oxidative stress curves were shown in Figure 3A,B. Compared with the CON group, the SOD activity of the lard and peanut oil groups decreased by 54.65% and 43.96%, respectively. A recent study showed that the decrease in SOD activity was associated with the accumulation of highly active free radicals, which caused the deterioration of cell membrane integrity and function [46]. Malondialdehyde (MDA) was the main active aldehyde produced by peroxidation, which was an indicator of tissue damage. Compared with the CON group, the MDA content of the lard oil group and the peanut oil group increased significantly. It indicated that lard and peanut oils can affect the generation and elimination of superoxide anion free radicals, and accelerate the formation rate of lipid peroxide by the peroxidation of unsaturated fatty acids in vivo. Therefore, this caused an imbalance in oxidation balance. However, there was a significant change in SOD and MDA concentrations in the CSO groups compared with the LA group. Supplementation with a higher amount of CSO (CSO-2, 50 mg/kg/BW) showed a more significant intervention on plasma lipid metabolism profiles in mice than supplementation with CSO-1 (30 mg/kg/BW). Those findings indicated that the hepatoprotective ability of CSO was related to its ability to reduce the free radical scavenging activities, thus stabilized the liver cell membrane.

### 3.5. Inflammatory Factors in the Hepatic Tissues

Oxidative stress and pro-inflammatory responses were produced in the body due to HFD-induced obesity, which resulted in decreased cytoplasmic nuclear factor kappa B (NF-ĸB) inhibitors. TNF-α could increase the permeability of vascular endothelial cells, activate neutrophils and lymphocytes, regulate the metabolic activity of other tissues, and promote synthesis. As a result, this could be employed as an indicator of oxidative stress, such as IL-1β, and TNF-α. As shown in (Figure 3C), peanut oil and lard improved expression levels of cytokines IL-1β, TNF-α, and INOS (*p* < 0.01). However, the CSO group had considerably lower levels of cytokines than the lard and peanut oil groups (*p* < 0.01). The data indicated that CSO could decrease the expression level of cytokines, and further confirmed the alleviation effect of CSO on HFD-induced liver inflammation in obese mice (Figure 3D). High fat diet caused metabolic, chronic, low-grade inflammation in the body because it induced obesity and the release of adipokines and inflammatory factors (For example, IL-1, IL-6, and TNF-α). Real time quantitative PCR (RT-qPCR) and western blotting were used to analyze the chronic and low-grade inflammation caused by obesity induced by a high-fat diet. Our results suggested that HFD raised the expression of cytokines IL-1β, TNF-α, and INOS (*p* < 0.01). Real time quantitative PCR (RT-qPCR) research revealed that, when compared with other oils, CSO significantly decreased the mRNA expression of these cytokines TNF-α (*p* < 0.01), IL-1β (*p* < 0.05), and INOS (*p* < 0.05) (Figure 3C). Changes in inducible nitric oxide synthase (INOS) levels before and after treatment, demonstrated that CSO considerably reduced the levels of plasma INOS and TNF-α, and help to improve the level of lipid metabolism. The CSO could reduce the protein expression of these cytokines, as evidenced by Western Blotting (WB) (Figure 3D). The results demonstrated that CSO could lower the cytokine expression level and alleviate liver inflammation in HFD-induced obese mice.

### 3.6. Histopathological Analysis

The detailed characterizations of histopathological changes in the HFD-induced liver were conducted by H&E staining (Figure 4). In the CON group, the liver tissues showed normal cytoplasm features, which were abundant and homogenous (Figure 4A). In contrast, the cytoplasm of the liver cells in the lard and peanut oil groups contained numerous spherical and tensioned lipid droplets of different sizes, especially in the lard group. The results suggested that mitochondria, endoplasmic reticulum, and the nucleus of adipocytes were under stress. Adipocyte volume increased that could be collapsed easily, leading to macrophage adipose tissue infiltration, and causing inflammatory responses. These symptoms were markedly alleviated in the CSO-treated groups. Hepatocyte morphology and structure were more compact and regular.

The hepatocyte structure of the CSO group was similar to that in the CON group, which was intact without degeneration and necrosis, being with uniform cytoplasm, and the nucleus was visible. Compared with the lard and peanut oil groups, the fat index volume of the CSO group was substantially reduced (Figure 1), and the average number of cells in a single field of view increased. Organ index results and histopathological analysis showed that the CSO improved hepatic cell lipidation, reduced liver fat deposition, and had no toxic side effects on organs.

## 4. Discussion

Lipid is essential for the balance of energy metabolism, tissue regeneration, and cell membrane stability. Inflammatory factors increased lipid metabolism disorders, while the changes in lipid metabolism also induced immune responses. As significant inflammatory agents, tumor necrosis factor-alpha (TNF-α) and interleukin-6 (IL-6) engaged in inflammatory responses and inhibited insulin signal transduction. TNF-α was a particular inflammatory cytokine, and IL-6 was generally produced by lymphocytes and monocyte macrophages, while the circulating IL-6 was mainly derived from adipose tissue, which was not only participating in the process of inflammation but also playing an essential role in regulating energy balance. It has been suggested that fatty acid-mediated dysfunction may increase insulin resistance and β-cell dysfunction. When fatty acid levels increased, glucose oxidation decreased. Fatty acids inhibited insulin signaling and led to a decrease in glucose uptake, glycogen synthesis, and glycolysis. Besides, fatty acids released from adipose tissue were also involved in the inflammatory process, therefore might cause the changes in NF-kB, IL-6, TNF-α, and TLRs signals. A recent study found that regulatory T (Treg) cells which were expressed the peroxisome-proliferator-activated receptor (PPAR-γ) were engaged in suppressing adipose tissue inflammation in obesity, which could be a target for treatment and prevention of adipose tissue inflammation and insulin resistance [10]. The deletion of PPAR-γ in mice increased the production of inflammatory cytokines, which was consistent with the liver inflammation as evidenced by the expression of TNF-α, IL-1 β, IL-6, IL-8, and the accumulation of TG in the liver.

From histopathological aspects, the mice fed with various dietary oils. The adipocytes had different degrees of volume changes in the liver. This led to inflammation of the adipose tissue. However, studies have shown that CSO inhibited the inflammatory response. With western blotting experiment, this idea was verified exactly.

The accumulation of lipid peroxidation products in cells decreased the normal biological activity of cells. Furthermore, it resulted in disorders in energy metabolism, cell signal transduction, and other dysfunctions [47,48]. In this study, CSO was found to significantly reduce body weight, serum TG, TC levels, fat deposition, regulate blood lipids, and improve lipid metabolism disorder in HFD mice (Figure 5). We found that the experimental results in the HFD mouse model were consistent with previous reports, which had been confirmed by previous clinical trials and animal experiments: a high fat diet could lead to an imbalance in lipid metabolism, and the polyunsaturated fatty acids in the body were attacked by ROS, resulting in lipid peroxidation and hyperlipidemia [49,50]. Our research also provided new evidence that CSO has a strong antioxidant capacity. For example, CSO protected the activity of antioxidant enzymes in vivo, effectively eliminated free radicals, maintained the balance between oxidation and antioxidant systems, and slowed down the degree of lipid peroxidation, protecting the body from the oxidative stress and preventing lipid metabolism disorder. According to reports, the coix seed has been served as nourishing food and used in traditional Chinese medicine for the treatment of inflammatory diseases, neuralgia, and neoplastic diseases [51]. Overall, CSO suppression played a critical role in obesity and dyslipidemia in high-fat diet-induced mice, and these effects were associated with an increase in MC-FA contents. These trials indicated the possibility of CSO as a food additive in the creation of novel functional oil.

## 5. Conclusions

Findings from this study have verified that CSO, which has a large amount of medium-chain fatty acids, is effective in preventing liver damage and fatty liver formation due to fat accumulation in HFD-induced obese mice. Compared with other dietary oils, testing liver index, fat index, serum lipid levels, relative expression of mRNA and so on, CSO successfully lowered triglycerides and total cholesterol, as well as the levels of INOS and TNF-α in serum lipid levels, then greatly improved lipid metabolism. Therefore, CSO has the potential to become a functional lipid that can prevent metabolic disorders in the future.

## Figures and Tables

**Figure 1 foods-11-03267-f001:**
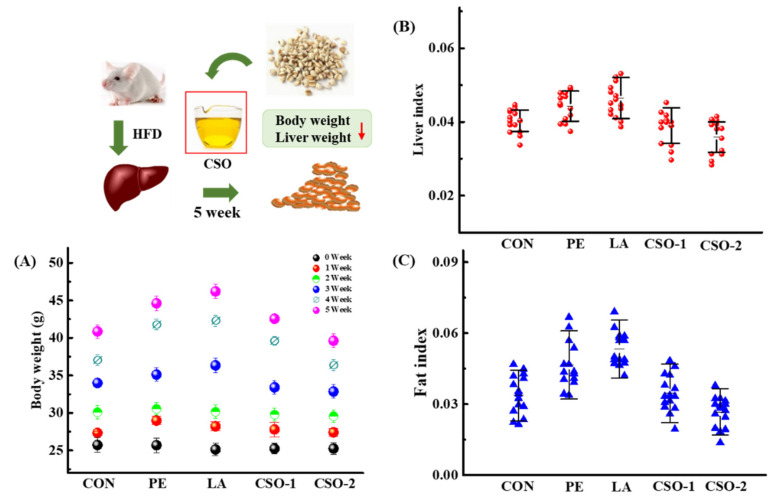
Effects of PE, LA and CSO therapy on the phenotype (body weight, liver, and fat index) in the HFD mice. A total of 75 HFD mice were randomly divided into control group (CON), PE, LA and CSO groups, and these mice from all groups were fed for 5 weeks. (**A**) Body weight alterations of mice during 5 weeks; (**B**) Liver index alterations from five groups; (**C**) Fat index alterations from five groups (CON: given normal saline group; PE: given peanut oil group; LA: given lard group; CSO-1 and CSO-2: CSO oral injection groups of 30 mg/kg/BW and 50 mg/kg/BW, respectively).

**Figure 2 foods-11-03267-f002:**
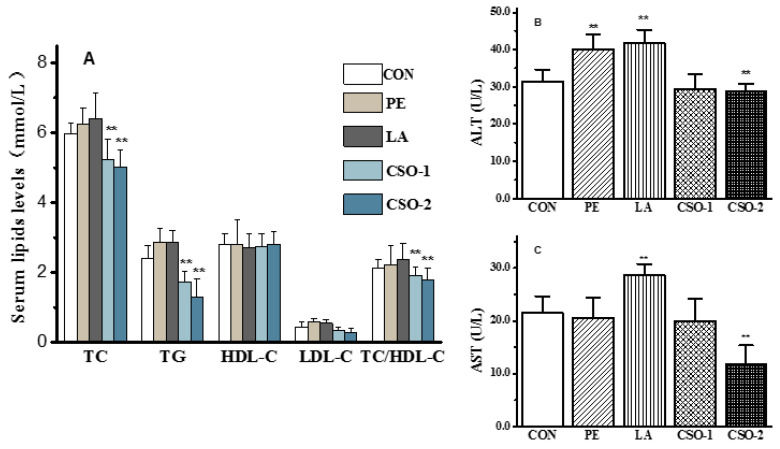
Effects of CSO on plasma lipids contents of HFD-induced obesity mice. (**A**) Serum lipids levels; (**B**) ALT; (**C**) AST. Data presented as means with the standard deviation (SD), *p* < 0.01 (**) compared to CON group (CON: given normal saline group; PE: peanut oil group; LA: lard group; CSO-1 and CSO-2: CSO oral injection groups of 30 mg/kg/BW and 50 mg/kg/BW, respectively).

**Figure 3 foods-11-03267-f003:**
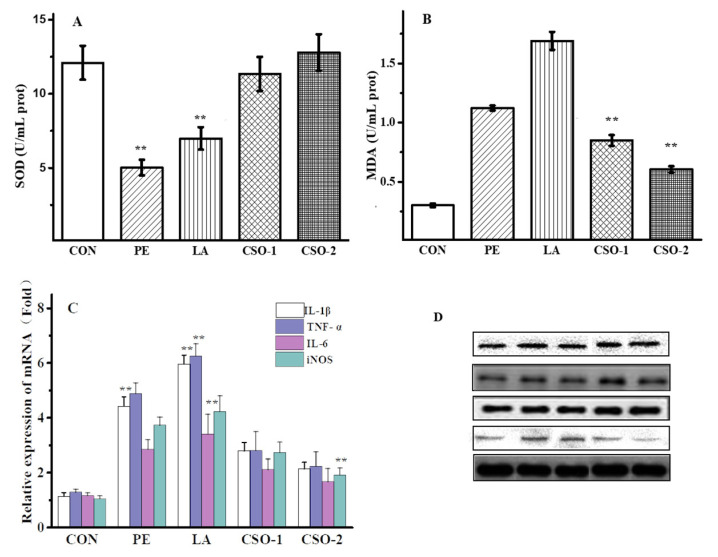
CSO inhibits the expression of oxidative stress and inflammatory factors in HFD-induced mouse liver tissue. (**A**) Effects of dietary fats on SOD level in the liver tissues of obesity mice; (**B**) Effects of CSO on MDA level in the liver tissues of obesity mice; (**C**) mRNA expression levels of cytokines were examined by RT-qPCR analysis (*n* = 15); (**D**) Protein expression levels of IL-6, TNF-α, and INOS were detected by Western blotting analysis (*n* = 3) *(p* < 0.01 (**) compared to the CON group) (CON: given normal saline group; PE: peanut oil group; LA: lard group; CSO-1 and CSO-2: CSO oral injection groups of 30 mg/kg/BW and 50 mg/kg/BW, respectively).

**Figure 4 foods-11-03267-f004:**
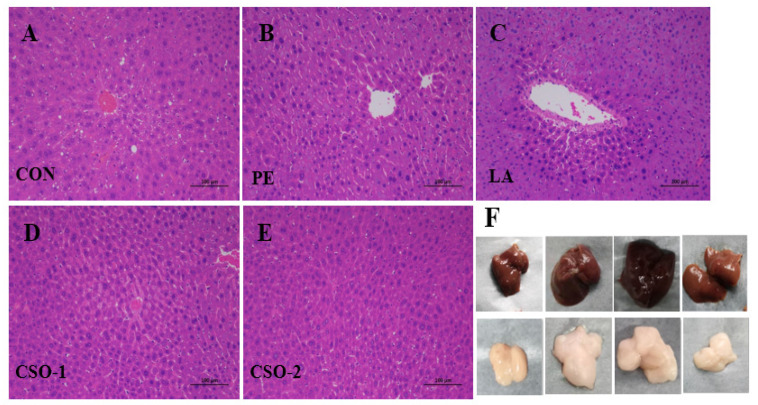
Morphological observation of mouse liver and epididymal fat. (**A**–**E**) Representative picture of pathological changes of liver tissues was observed by H&E staining (200-fold magnification); (**F**) Representative photos of mice livers and epididymal fat pads from control group, PE group, LA group and CSO groups (CON: given normal saline group; PE: peanut oil group; LA: lard group; CSO-1 and CSO-2: CSO oral injection groups of 30 mg/kg/BW and 50 mg/kg/BW, respectively).

**Figure 5 foods-11-03267-f005:**
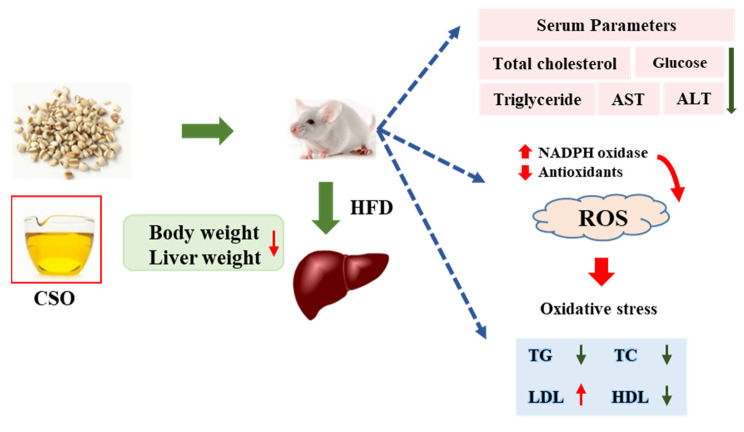
Schematic representation of the mechanism of action through which CSO affects obesity induced by a high-fat diet in mice.

**Table 1 foods-11-03267-t001:** Component and distribution of fatty acids in CSO (mol%).

	Peanut Oil (PE)			Lard (LA)			Coix Seed Oil (CSO)		
Fatty Acid	Total Content (%)	sn-2	sn-3	Total Content (%)	sn-2	sn-3	Total Content (%)	sn-2	sn-3
C8:0	0.41	0.03	0.38	0.21	0.17	0.04	3.41	0.43	2.98
C10:0	6.28	3.51	2.77	4.28	2.15	2.13	45.28 *	24.5	20.78
C12:0	10.21	8.24	1.97	8.71	6.34	2.37	42.21 *	22.24	19.97
C14:0	21.22 *	12.23	8.99	12.62	10.45	2.17	2.22	1.23	0.99
C16:0	28.03	17.55	10.48	30.03 *	11.45	18.58	1.03	0.45	0.58
C18:0	31.67 *	16.32	15.35	38.67 *	19.32	19.35	0.67	0.32	0.35
9C18:1	0.63	0.32	0.31	0.91	0.32	0.59	3.21	1.32	1.89
11C18:1	0.23	0.12	0.11	1.13	1.02	0.11	0.23	0.02	0.21
9c12cC18:2	0.44	0.12	0.32	0.58	0.32	0.26	1.48	0.82	0.66
9t12c15tC18:3	0.91	0.43	0.48	0.92	0.33	0.59	0.93	0.53	0.4
9c12c15tC18:3	0.33	0.15	0.18	0.76	0.15	0.61	1.13	0.93	0.2

Differences were considered significant at *p* < 0.05 (*) compared to CON group.

## Data Availability

Data is contained within the article.

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
