# Peer review of "Effects of Coix Seed Oil on High Fat Diet-Induced Obesity and Dyslipidemia"

_foods, 2022, doi:10.3390/foods11203267_

Round 1

Reviewer 1 Report

1- in page 1 line17 the standard deviation should added for the amounts of capric acid and lauric acid being 35.28% and 22.21%

2- the aim of the work should be rewrite in clear to easy understand  in page 2 line 64-70 

3- in page 4  in table 1 till  line 170 what is the  meaning of PE &LA . Author should add all appreviation under table and figures for example (figure 1, a, b,c )

4- the english need  to improved

4- 

Reviewer 2 Report

General Comment:

The manuscript is well written. The authors investigated “Effects of Coix Seed Oil and High Fat Diet-Induced Obesity and Dyslipidemia”. The study is interesting and adds to the existing body of knowledge. Some errors need clarification and revisions. 

Detail’s comment:

1.     Please change the “in-text” reference format according to the MDPI reference format in the whole manuscript (Introduction to Discussion). “[citation number]”

2.     Page 1, Line 42: “TNF-a

3.     Page 2, Line 47: commonly known as “adlay” [16], is a potential food source.

4.     Page 2, Line 74-80: Please add catalog number for assay kit and antibodies

5.     Page 3, Line 123-127: Please add catalog number and manufacturer for assay kits

6.     Page 3, Line 129-133: Please describe details of the histopathological process (Blocking of samples, cutting of samples-include thickness of the sample, dehydration with series of alcohol, etc..). Please state the magnification of the photograph captured. 

7.     Page 3, Line 139-139: Please add the catalog number for the assay kit

8.     Page 3, Line 140: Please add catalog number for antibodies

9.     Page 6, Line 224: Figure 2B, CSO-2 vs CON significantly (**p < 0.01) reduce ALT? How about CSO-1 vs CON? Any significant difference? The means ALT values of CSO-1 and CSO-2 look equal. Please justify. 

10.  Page 8, Line 292: Please label or put an arrow for central vein, hemorrhage, and necrosis in the figure for better understanding. 

Reviewer 3 Report

The manuscript should be carefully revised based on the major following comments:

1.         The language needs to be edited throughout the text. There are obvious grammatical, punctual, and syntax errors in the abstract and text.

2.         All the abbreviations except for CSO and HFD are additional and should be deleted because they were not repeated.

3.         In the abstract, ‘capric acid and lauric acid’ should be ‘capric and lauric acids’. Moreover, there is no necessity to mention ‘with a high con- 17 centration of medium chain fatty acids’ because it was earlier stated. Accordingly, the authors should add other significant numeric results to the abstract. It suffers from low data and should be enriched with more comprehensive information.

4.         The number of references in this manuscript is very low, while the authors should use the potential of adding more keywords (up to 10) for more visibility of other work after publication. In addition, more keywords related to obesity, and related health status/disorders.

5.         Space should be put between two terms. This error can be frequently observed in the text. For example, line 40, superoxide dismutase(SOD). Also, check the symbols such as TNF-a (line 42; it should be alpha), Check the text for these error types carefully.

6.         The introduction is good but there is a gap in this section. It is a literature review on the similar effects of other edible oils. This should be carefully reviewed and related references should be added. Then, the authors can make studying gaps and point out the aims of performing this study.

7.         In the part of experimental, the authors should explain what was the used extraction technique to obtain the oil (for example, cold press or solvent extraction) If the solvent was the applied method to extract the oil, the authors should mention the solvent type and extract method (e.g., Soxhlet)

8.         It seems that how the use of references in the text is wrong and is not according to the MDPI journals.

9.         Line 75: aspartate transaminase (AST,) > delete coma in the parenthesis. Line 77, it should be β-actin,.

10.      All other reagents were used to analyze the highest commercial grade available (line 79) should be mentioned that they were provided by Merck and Sigma Chemical Co. to avoid bringing the name of two companies in other parts of the section of Experimental

11.      Equations should have a number to follow up easily.

12.      The information related to name, model, manufacturing company, city, and country should be given for all tools and equipment applied in your work.

13.      Subsections 2.4, 2.5, and 2.6 should have at least a reference. Otherwise, the authors should mention the details of the applied methodologies.

14.      Results were not well discussed in the manuscript. All discussions given in the section of Discussion should be included in the Results and then the authors and readers can realize some findings were not discussed. Also, if you were going to discriminate Results from Discussion, there are some references in the Results. More discussions should be presented related to the molecular and biochemical mechanisms of the CSO effects. Most sentences given in the Discussion are self-explanatory and should have a reference(s). Also, the lack of discussion should be compensated by adding more recent related references for more comprehensive discussions.

15.      The quality of Figures 3 (B and D) should be increased!

16.      The manuscript has no conclusions section!

17.      The whole references are not in the format of Foods or MDPI journals. As already mentioned, the number is very low and should be increased significantly. Most references were given in the introduction and this is frustrating.

18.      The ethical aspects were mentioned in the text and should be mentioned at the end of the manuscript before bringing the part of References

Round 2

Reviewer 3 Report

There are still some comments to improve the content as follows:

1. The language still should be improved! Please check for more readability. Just for one note, in line 325, Tumor should be ''tumor''. These mistakes are a lot in the text.

2. The final references still are not under the journal's guideline. All the words in the title of cited papers should be written with an uppercase letter in the initial. Journals' names cited in the manuscripts should be abbreviated, while there are still some whole names. Scientific names should be written in italic. Check other instructions carefully.

3. Keywords still are four and there are no additional keywords for doing my comments in the first revision.

4. The section of the Conclusion is a discussion! This comment also was not implemented. Many sentences in this section should be transferred to the previous section. Basically, we can not have any figure in the conclusion and this figure should be transferred to the discussion and presented explanations there! The conclusion should not have any reference.

5. Personally, I still believe that more discussions should be added to the results obtained from this research. 

6. Once again, I EMPHASIZED that the authors must not use the abbreviated format in the initial of the sentence.

7. In Table 1, it is not essential to give a footnote for ''PE, LA, and CSO''. Give the whole names in the first raw and deleted this footnote. The use of Table 1 in the footnote is wrong!
